# Pain prevalence in hospitalized patients at a tertiary academic medical center: Exploring severe persistent pain

Chieh-Liang Wu[1,2,3], Yin-Lurn Hung[3,4], Yan-Ru Wang[5], Hui-Mei Huang[6], Chia-Hui Chang[6], Chih-Cheng Wu[7], Chih-Jen Hung[7], Te-Feng Yeh[5]*

1 Department of Critical Care Medicine, Taichung Veterans General Hospital, Taichung, Taiwan, 2 Department of Automatic Control Engineering, Feng Chia University, Taichung, Taiwan, 3 Center of Quality Management, Taichung Veterans General Hospital, Taichung, Taiwan, 4 Department of Administration, Taichung Veterans General Hospital, Taichung, Taiwan, 5 Department of Healthcare Administration, Central Taiwan University of Science and Technology, Taichung, Taiwan, 6 Department of Nursing, Taichung Veterans General Hospital, Taichung, Taiwan, 7 Department of Anesthesiology, Taichung Veterans General Hospital, Taichung, Taiwan

* tfyeh@ctust.edu.tw

**Data Availability Statement:** All relevant data are within the manuscript and its Supporting Information files.

**Funding:** The author(s) received no specific funding for this work

## Abstract

### Objective

The pain prevalence of inpatients is not a well-studied medical issue in Asia. We have aimed to evaluate pain prevalence and characterize those patients who have suffered from severe, persistent pain.

### Methods

We investigated pain prevalence using a quota sampling from 19 general wards during the year 2018. Using a structured questionnaire, eight interviewers visited patients at an age ≥ 20 years, and who had been staying in general wards for ≥ 3 days. Those patients were excluded if they were unable to respond to the interview questions. If they reported pain during hospitalization, the maximum pain level and the duration of pain suffered in the past 24 hours were assessed. Care-related pain was also surveyed.

### Results

A total of 1,034 patients (M/F, 537/497) completed the survey. Amongst them, 719 patients (69.5%) experienced pain, with moderate and severe pain levels being 27.3% and 43%, respectively. Surgery was considered as it related to pain, including significantly severe pain. The top 3 care-related pain causes were needle pain, wound dressing, and change in position/chest percussion. Change in position/chest percussion and rehabilitation were associated with severe, persistent pain.

**Competing interests:** The authors have declared that no competing interests exist.

## Conclusions

Pain is common in approximately 70% of inpatients, with surgery being associated with severe pain. Mobilization and rehabilitation may lead to severe, persistent pain. The periodic study of pain prevalence is essential in order to provide precise pain management.

## Introduction

Pain is a common symptom experienced by hospitalized patients. The intent of the "Pain is the fifth vital sign" campaign (Presidential Address to the American Pain Society, 1996, Campbell) was to encourage both doctors and nurses to listen to their patients as they assess their pain [1]. If we want to become a "Towards a Pain-Free Hospital," we have to monitor the pain levels of all hospitalized patients and assess the adequacy of the pain being treated.

In the United States, the Hospital Consumer Assessment of Healthcare Providers and Systems (HCAHPS) survey consists of a battery of questions that measure ten core concepts, including pain management. This survey is administered to a random sample of adult patients between 48 hours and 6 weeks after discharge. Nearly 30% of inpatients suffered from pain during hospitalization in 2012 [2]. In Europe, several reports used questionnaires and the numeric rating scale (NRS) to investigate the pain prevalence of hospitalized patients (N = 526~4,523). Approximately 52–89.5% of the patients had experienced pain (NRS>3) during the previous 24 hours [2–9]. During a one-day cross-sectional survey of pain prevalence, 59% (554/938) of patients reported pain within the previous 24 hours, while 58% (540/938) had experienced care-related pain within the previous 15 days. Care-related pain happened not only in the internal medicine units but also in the surgical units [10]. In Asia, Xiao et al. conducted a 9-week structured and systemic interview to determine the pain prevalence amongst 3,248 patients in a Chinese hospital. The pain prevalence was measured at 63.4%, with more than 97% of the patients suffering from moderate to severe pain [11]. In Taiwan, Tsai et al. study showed that pain prevalence was at 50% among the community-dwelling elderly, but not for the inpatients [12].

Studies regarding the pain prevalence of hospitalized patients in Asia are limited, and there is no data at all available in Taiwan. Because of our commitment to the project "Towards a Pain-Free Hospital," we conducted a prospective study to evaluate the prevalence of pain amongst hospitalized patients, and characterized those patients who suffered from severe and persistent pain. The causes of care-related pain were also surveyed.

## Method

### Study setting and approval

Taichung Veterans General Hospital (TCVGH) is a 1,500-bed teaching hospital and tertiary referral medical center in central Taiwan. TCVGH contains 19 general wards for adults, and handles approximately 50,000 hospitalized patients each year, while performing around 40,000 operations. The cohort study, conducted from August 21 to October 31 in 2018, surveyed the pain prevalence of hospitalized patients prospectively. The Institutional Review Board I & II of Taichung Veterans General Hospital approved the study protocol (Protocol no./IRB, TCVGH No: CE18236B).

## Questionnaire

After an extensive review of the available literature, our pain care specialists (two physicians, two nurses, and one statistician) developed the questionnaire. It surveys the prevalence of patients' pain during hospitalization, along with their experience while receiving pain management from our physicians and nurses. The questionnaire is composed of 36 questions, divided into 4 main parts. The first part asks for demographic data, including age, gender, education, weight, disease categories, and surgical history. The second part is for evaluating pain status, the types of care-related pain, and a patient's literacy regarding pain. The third part is to define their experience surrounding pain management from the physicians and nurses. And the final part is to measure the patient's overall satisfaction. Five experts reviewed the questionnaire with a Content Validity Index (CVI) of 0.90. The study included the results of part 1 and 2 of the questionnaire.

The current report focused on the prevalence of pain, including its severity and duration, while also considering care-related pain. In the questionnaire, pain was self-evaluated by the patients using the NRS, with "no pain: 0" and "worst possible pain: 10" rating scale. Initially, we asked the patient whether they had experienced pain or not during hospitalization. If the patient said "yes", both their maximum pain and duration of the pain suffered in the past 24 hours ("worst pain over 24 hours") were assessed. We also listed the common factors related to care-related pain, including needle pain, wound dressing pain, change posture/chest percussion, nasogastric tube, foley catheter, rehabilitation exercise, chest tube, and other drainage tubes. The patients labeled all the items of care which were causing their pain. Based on our standard of operation of "Toward a Pain-Free Hospital", we have to respond to the breakthrough pain within one hour and then define the duration of pain longer than 4 hours is persistent pain.

## Sampling and interview

The patients were enrolled if they were older than 20 years, and had stayed in the general ward for at least 3 days. Patients were excluded if they were critically ill, had lost their consciousness, or were unable to respond to the interview due to their underlying diseases. To recruit hospital-wide patients, we conducted a quota sampling from each general ward. The number of beds of each general ward was the quota and we stopped enrollment if the quota was reached. Prior to starting data collection, we trained eight interviewers in order to reduce the bias of collecting information. When the interviewers arrived at the wards, the head nurse of each ward listed the patients who had met the enrollment criteria. The interviewers then sampled the patients randomly, and performed interviews until the quota of each ward had been reached. After the patients signed the informed consent document, they began the interview and stopped at any time if the patient refused to continue, or were too sick to answer any more questions. The patients answered the questionnaire themselves. The interviewers imputed the patients' answers into the web-based questionnaire site via a mobile IPAD.

## Statistics

Patient information was anonymized and de-identified prior to analysis. We defined the levels of pain using the NRS as mild (1–3), moderate (4–6), and severe (7–10). We listed the demographic data, including gender, age, weight, education, disease categories, and surgical history. We used the Chi-square test and multiple logistic regression to identify the factors related to pain, including severe pain, among all the patients. We also identified the factors related to moderate to severe pain, and pain lasting longer than 4 hours amongst those patients experiencing pain. The factors surrounding care-related pain were listed from the top

frequency, and analyzed the factors related to severe pain and pain lasting longer than 4 hours. Statistical significance was assumed for an alpha level of $p < 0.05$. All the analysis was performed using SPSS Windows Version 21.0.

## Results

### Demographic data and factors related to pain

A total of 1,034 patients completed the survey. The demographic data is shown in Table 1. Amongst them, 719 patients (69.5%) experienced pain during their hospitalization, while 312 (30.2%) experienced severe pain (NRS 7–10). The percentages of pain were 73.2% in young adults (age $\leq$ 39 years old), and 57.1% in the elderly (age $\geq$ 80 years old). Data also showed a decreasing trend with each 10-year increment of age. Using logistic regression, we noted that those patients older than 80 years of age had fewer pain problems (Odds ratio: 0.488, p<0.01). Those patients cared for in the Department of Internal Medicine also suffered less pain (Odds ratio: 0.368, p<0.001). However, surgery is a strong factor related to pain (Odds ratio: 3.401, p<0.001), and also severe pain (Odds ratio: 1.5, p<0.01).

### Pain severity and duration

Seven hundred and nineteen patients (69.5%) had experienced pain during their hospitalization. The severity and duration of their pain within the latest 24 hours were reported by themselves. Using a numeric rating scale, we divided those patients experiencing pain during hospitalization into 4 groups no pain (0), mild pain (1–3), moderate pain (4–6), and severe pain (7–10). The percentage of moderate and severe pain was 27.3% and 43%, respectively (Fig 1). This meant that more than 70% of patients had suffered from moderate to severe pain within the last 24 hours (Fig 1). We also classified the duration of pain into 4 groups: transient, less than 1 hour, 1–4 hours, and longer than 4 hours. The percentage of patients who suffered from persistent pain for 1 to 4 hours was 14.7% while those suffering longer than 4 hours was 24.6% (Fig 2). The factors related to moderate to severe pain (NRS 4–10) were explored. As we expected, surgery was significantly related to moderate to severe pain, but was not related to pain involving a more prolonged duration period of over 4 hours (Table 2). Only patients within the age range of 70~79 were associated with severe and persistent pain in comparison to young patients (age $\leq$ 39). (Odds ratio 2.563, p<0.05)

### Types of care-related pain

Of the 503 patients who reported care-related pain, 212 patients reported one type of care-related pain, while 291 patients reported two or more types of care-related pain. The three most frequently reported care-related pain types were needle pain (n = 353; 44.2%), wound dressing pain (n = 177; 22.2%), and change in position/chest percussion (n = 139;17.4%) (Table 3). Interestingly, wound dressing pain was severe but not persistent. Change in posture/chest percussion and rehabilitation exercise were positively associated with both severe and persistent pain.

## Discussion

We can proceed forward with the goal of reaching a "Towards a Pain-Free Hospital" from the prospective survey of pain prevalence among hospitalized patients. Those patients who underwent an operation and were cared for at the surgical service level were more likely to report their pain as being moderate to severe. However, pain related to surgery did not last for a long period of time. In contrast, a change in position/chest percussion and rehabilitation exercise

**Table 1. Demographic data and risk factors for pain and severe pain.**

| Characteristics | ALL[a] | With Pain | Severe pain |
|---|---|---|---|
| | N = 1,034 (%) | N = 719 (%) | N = 312 (%) |
| **Gender** | | | |
| Male | 537 (48.1) | 353 (49.1) | 155 (49.7) |
| Female | 497 (51.9) | 366 (50.9) | 157 (50.3) |
| **Age (years old)** | | | |
| ≤39 | 168 (16.2) | 123 (17.1) | 51 (16.3) |
| 40~49 | 121 (11.7) | 88 (12.2) | 38 (12.2) |
| 50~59 | 240 (23.2) | 174 (24.2) | 75 (24.0) |
| 60~69 | 257 (24.9) | 175 (24.3) | 77 (24.7) |
| 70~79 | 136 (13.2) | 95 (13.2) | 49 (15.7) |
| ≥80 | 112 (10.8) | 64 (8.9)[b] | 22 (7.1) |
| **Weight (Kg)** | | | |
| ≤50 | 144 (13.9) | 102 (14.2) | 45 (14.4) |
| 50~59 | 277 (26.8) | 194 (27.0) | 81 (26.0) |
| 60~69 | 322 (31.1) | 223 (31.0) | 93 (29.8) |
| 70~79 | 186 (18.0) | 120 (16.7) | 56 (17.9) |
| ≥80 | 105 (10.2) | 80 (11.1) | 37 (11.9) |
| **Education** | | | |
| Elementary school | 220 (21.3) | 143 (19.9) | 65 (20.8) |
| Junior high school | 159 (15.4) | 108 (15.0) | 45 (14.4) |
| Senior high school | 284 (27.5) | 211 (29.3) | 93 (29.8) |
| Bachelor's degree or more | 299 (28.9) | 211 (29.3) | 89 (28.5) |
| Illiterate | 72 (7.0) | 46 (6.4) | 20 (6.4) |
| **Marriage** | | | |
| Unmarried | 142 (13.7) | 100 (13.9) | 43 (13.8) |
| Married | 765 (74.0) | 543 (75.5) | 232 (74.4) |
| Other | 127 (12.3) | 76 (10.6) | 37 (11.9) |
| **Disease categories** | | | |
| Surgery | 400 (37.8) | 320 (44.5) | 158 (50.6) |
| Head, Neck, Ophthalmology | 55 (5.3) | 40 (5.6) | 23 (7.4) |
| Internal Medicine | 509 (49.2) | 303 (42.1)[b] | 112 (35.9)[b] |
| Gynecology | 70 (6.8) | 56 (7.8) | 19 (6.1)[b] |
| **Operation** | | | |
| No operation | 490 (52.6) | 276 (38.4) | 103 (33.0) |
| With operation | 544 (47.4) | 443 (61.6)[b] | 209 (67.0)[b] |

N (%): Data are expressed as the case number and its percentage among total cases.

[a] three cases had missing data of pain.

[b]: Odds ratio of multiple logistic regression for pain or severe pain significantly (p< 0.05).

were associated with both severe pain and a long duration. We have to pay attention to the causes of severe persistent pain in order to provide precise pain management amongst patients.

## Pain prevalence in a medical center

We conducted a similar survey of pain prevalence to the one performed by Salomon et al. at a French teaching hospital in 2002 [6]. They reported that 55% of 998 inpatients experienced pain. In a German university teaching hospital, medical staff reported pain prevalence in 63%

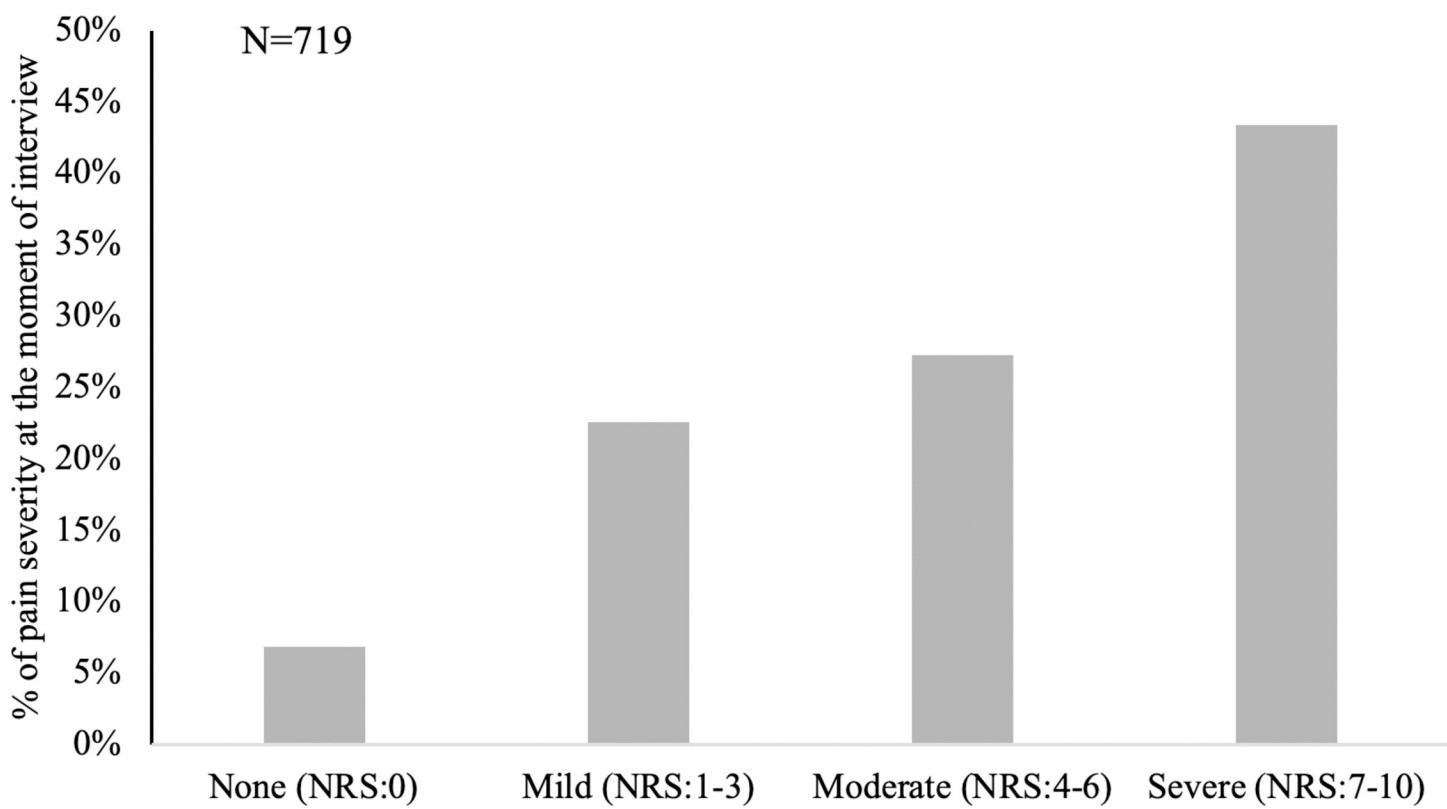

**Fig 1. Pain severity at the moment of the interview.** Seven hundred and nineteen patients had experienced pain within the last 24 hours when responding to the questionnaire. The pain severity at the moment of the interview was divided into 4 grades using a Numerical Rating Scale (NRS): none (0), mild (1–3), moderate (4–6), and severe pain (7–10). The percentage of each group is shown.

of patients during the 24 hours preceding the interview. They also monitored pain levels both at rest and during movement as being 33% and 50% respectively, at the time of the interview [7]. Melotti et al. from Italy showed that pain prevalence occurred in 52% of patients during the 24 hours previous to their interview [4]. Another study involving pain prevalence came from Sweden and disclosed that 65% (494/759) of patients had experienced pain during the past 24 hours [13]. Fabbian et al. from Italy conducted a 6-month prospective observational study, which showed 63% of patients with significant pain (NRS $\geq$ 3), and 7.6% with severe pain (NRS $\geq$ 7) amongst the consecutively enrolled 526 patients within their internal medicine department [9]. The prevalence of pain in our study was 69.5%, which was similar to the above studies. However, we noted that the percentage of the patients with severe pain was up to 30% (312/1034), which was much higher than the findings of Rabbian et al [9]. Our hospital is a 1500-bed tertiary academic medical center and take care the patients with high severity (Case Mixed index 1.3 in 2018). Nearly half of the enrolled patients received surgical intervention (Table 1). Therefore, high percentage of severe pain seems rational in the study.

The prevalence of pain possibly varied due to the sample methods and study designs used. Two types of study design were noted. The first was an exhaustive cross-sectional study, much like a "snapshot" of pain prevalence amongst the hospitalized patients on an index day, as performed by Salomon et al., Wadensten et al., and Melotti et al [4, 6, 13]. The second was the interviewing of patients ward by ward, which required one to two months to complete during a hospitalized survey of pain prevalence. Our study method implemented the second type, and was similar to that performed by Strohbuecker et al [7]. In terms of the questionnaire, all studies

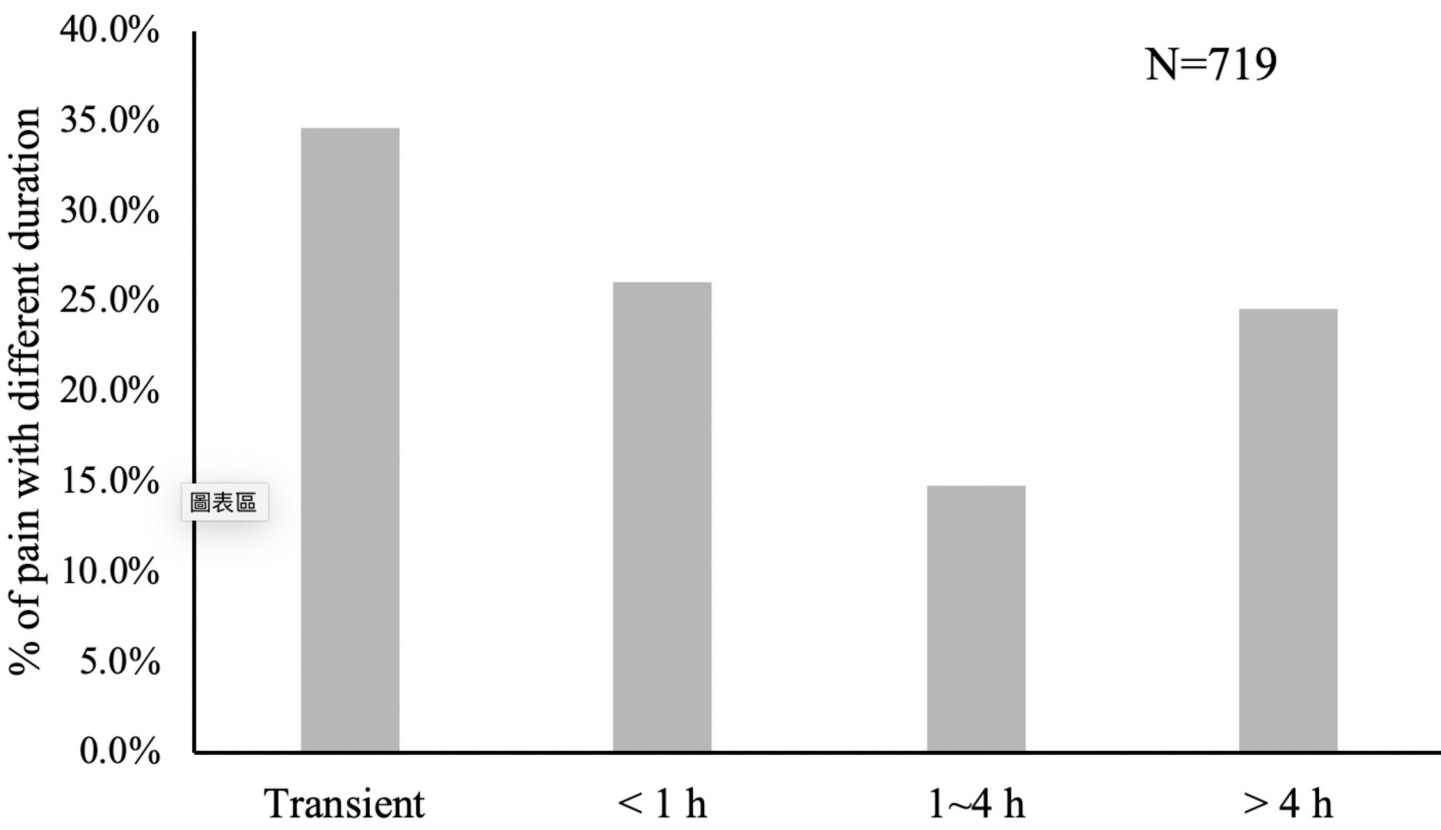

**Fig 2. Severe and persistent pain.** Seven hundred and nineteen patients had experienced pain within the last 24 hours when responding to the questionnaire. The duration of pain within the last 24 hours was divided into 4 groups titled: transient, < 1 hour, 1–4 hours, and > 4 hours. The percentage of each group is shown.

investigated each patients pain experience during the past 24 hours previous to the interview. Those studies usually excluded pediatric, obstetric, and intensive care units. No matter the different study designs, we noted that the pain prevalence was approximately 50~70%, which denoted that the pain amongst the inpatients was both frequent and severe. One of the studies from Italy disclosed an extremely high pain prevalence of up to 91.2% (3,864/3,931). However, this study had some weaknesses in its study design. It recruited 20 small or medium-sized hospitals on a voluntary basis, with the type of questionnaire being used contributing to a slight overestimation of the prevalence of pain. This study did not provide the setting of a large teaching hospital.

## Gender and pain prevalence

The prevalence of pain was nearly the same between males and females in our study. Some literature has shown that the prevalence of pain is higher in female than male patients. Female vs. male percentages were: 67% vs. 58%, 72% vs. 58%, and 88% vs. 78% in the studies of Melotti et al., Wadebsteb et al., and Zoëga et al. respectively [4, 8, 13]. However, some studies did not relate the risk of suffering pain to gender predominance [6, 7, 9]. Whether gender is a significant risk factor to developing pain remains controversial. Nevertheless, we should continue to pay attention to pain conditions equally in both male and female patients.

## Age and pain prevalence

Using logistic regression, we noted that those patients older than 80 years of age experienced fewer pain problems. Some studies also showed no age-related predominance of pain prevalence

**Table 2. Risk factors of pain (NRS 4–10), duration $\geq$ 4 h and both.**

| Characteristics | ALL N (%) | NRS 4–10 N (%) | Pain $\geq$ 4 h N (%) | NRS 4–10 & Pain $\geq$ 4 h N (%) |
|---|---|---|---|---|
| Case number | N = 719 | N = 508 | N = 177 | N = 143 |
| **Gender** | | | | |
| Male | 353 (49.1) | 249 (49.0) | 89 (50.3) | 71 (49.7) |
| Female | 366 (50.9) | 259 (51.0) | 88 (49.7) | 72 (50.3) |
| **Age (years old)** | | | | |
| $\leq$39 | 123 (17.1) | 90 (17.7) | 19 (10.7) | 13 (9.1) |
| 40~49 | 88 (12.2) | 67 (13.2) | 21 (11.9) | 19 (13.3) |
| 50~59 | 174 (24.2) | 120 (23.6) | 42 (23.7) | 30 (21.0) |
| 60~69 | 175 (24.3) | 126 (24.8) | 53 (29.9)[a] | 44 (30.8) |
| 70~79 | 95 (13.2) | 68 (13.4) | 29 (16.4) | 28 (19.6)[a] |
| $\geq$80 | 64 (8.9) | 37 (7.3)[a] | 13 (7.3) | 9 (6.3) |
| **Weight (Kg)** | | | | |
| $\leq$50 | 102 (14.2) | 69 (13.6) | 21(11.9) | 16(11.2) |
| 50~59 | 194 (27.0) | 140 (27.6) | 54(30.5) | 48(33.6) |
| 60~69 | 223 (31.0) | 152 (29.9) | 61(34.5) | 48(33.6) |
| 70~79 | 120 (16.7) | 88 (17.3) | 24(13.6) | 17(11.9) |
| $\geq$80 | 80 (11.1) | 59 (11.6) | 17(9.6) | 14(9.8) |
| **Education** | | | | |
| Elementary school | 143 (19.9) | 101 (19.9) | 39 (11.9) | 35 (24.5) |
| Junior high school | 108 (15.0) | 83 (16.3) | 30 (30.5) | 28 (19.6) |
| Senior high school | 211 (29.3) | 147 (28.9) | 51 (34.5) | 38 (26.6) |
| Bachelor's degree or more | 211 (29.3) | 147 (28.9) | 43 (13.6) | 31 (21.7) |
| Illiterate | 46 (6.4) | 30 (5.9) | 14 (9.6) | 11 (7.7) |
| **Marriage** | | | | |
| Unmarried | 100 (13.9) | 72 (14.2) | 20 (11.3) | 15 (10.5) |
| Married | 543 (75.5) | 383 (75.4) | 129 (72.9) | 105 (73.4) |
| Other | 76. (10.6) | 53 (10.4) | 28 (15.8) | 23 (16.1) |
| **Disease categories** | | | | |
| Surgery | 320 (44.5) | 249 (49.0) | 81 (45.8) | 70 (49.0) |
| Head, Neck, Ophthalmology | 40 (5.6) | 28 (5.5) | 9 (5.1) | 6 (4.2) |
| Internal Medicine | 303 (42.1) | 194 (38.2)[a] | 75 (42.2) | 59 (41.3) |
| Gynecology | 56 (7.8) | 37 (7.3) | 12 (6.8) | 8 (5.6) |
| **Operation** | | | | |
| No operation | 276 (38.4) | 177 (34.8) | 66 (37.3) | 52 (36.4) |
| With operation | 443 (61.6) | 331 (65.2)[a] | 111 (62.7) | 91 (63.6) |

N (%): Data are expressed as the case number and its percentage among total cases.

[a]: Odds ratio of multiple logistic regression for with pain or severe pain (p< 0.05).

[6, 7], however they concluded that age was an essential factor in pain prevalence. Fabbian et al. revealed that age was an independent factor associated with pain [8, 9] and that pain prevalence was higher in the age group of 20~40 years, but decreased gradually as age increased [9]. Melotti et al. disclosed that pain prevalence was high amongst young adults [4]. A research study from Jerusalem regarding the epidemiology of chronic pain coinciding with advancing age, showed the prevalence of pain to be 73% at ages 70~71, 81% at ages 77~78, 56.3% at ages 85~86, and 31.2% at ages 90~91 [14]. The prevalence of experiencing chronic pain of at least 3 months, over a period of the past 6 months in Singapore was 8.7%, higher in females (10.9%), and increased

**Table 3. Risk factors of moderate to severe pain, pain longer than 4 hours and both, in reported care-related pain.**

| Care-related pain | Frequency of reported care-related pain N = 799 (%) | NRS 4–10 N = 508 (%) | Pain ≥ 4 h N = 312 (%) | NRS 4–10 & Pain ≥ 4 h N = 177 (%) |
|---|---|---|---|---|
| Needle Pain | 44.2 | 49.3 | 55.4 | 55.2 |
| Wound dressing pain | 22.2 | 29.0 [a] | 28.2 | 30.1 |
| Change posture/Chest percussion | 17.4 | 21.9 [a] | 28.2 [a] | 32.2 [a] |
| Nasogastric tube | 4.8 | 5.3 | 6.8 | 7 |
| Foley Catheter | 4.4 | 4.1 | 8.5 | 6.3 |
| Rehabilitation | 3.6 | 4.3 | 7.3 [a] | 7.0 [a] |
| Central venous catheter | 1.6 | 2.2 | 1.1 | 1.4 |
| Chest tube | 1.4 | 1.8 | 1.7 | 2.1 |
| Drainage tube | 0.5 | 0.4 | 0 | 0 |

[a]: p< 0.05 by Chi-square test.

with age [15]. Another study showed that the prevalence of pain was 67.3% amongst elderly hospitalized patients at the age of 78 ± 8.1 [16]. The pain prevalence in geriatric patients was similar to our study (69.8% at ages 70~79). Poor self-rated health was more common in advanced aged patients [14]. The geriatric population may have limited ability to express their pain. According to the above evidence, pain prevalence in geriatric patients remains high, although the percentage of pain prevalence decreased with an increase in age. Therefore, we should not underestimate or minimize pain problems within the elderly population.

## Chronic pain may superimpose on the pain of inpatients

Inpatients may have suffered from chronic pain (lasting for 3 months or more) prior to hospitalization. Up to 21~ 44% of inpatients suffered from continuous chronic pain in the 24 hours prior to being interviewed [4, 6]. In our study, we did not investigate how many patients were experiencing chronic pain, which involves pain during the period before to after hospitalization. We only attempted to identify the duration of non-relief pain during hospitalization. The results were severe due to the fact that up to 70% of the inpatients experienced moderate and severe (NRS 4~10) pain, with 24.6% of them suffering from pain longer than 4 hours. We found only those in the age range of 70~79 suffered from severe pain over a long duration. Further studies are warranted in order to explore chronic pain and its impact on pain prevalence in our hospital.

## Risk factors related to severe persistent pain in hospitalized patients

Care-related pain has been well studied. The top 3 procedures resulting in care-related pain were vascular puncture, mobilization, and other invasive procedures or therapeutic care [10, 17]. Based upon our clinical practice, we designed the items associated with care-related pain to be easily understood by our patients, and then counted the frequency of care-related pain. The top cause of care-related pain, vascular puncture, was the same as found in the above-mentioned literature. We also noted that wound dressing pain was severe but not persistent. This result was consistent with the finding that surgery was associated with severe pain, but not over a long duration (Table 2). Unexpectedly, both change in posture/chest percussion and rehabilitation were definite risk factors regarding severe pain over a long duration. As our authors observed in our hospital, this scenario was common for "old age" (70~79) patients with multiple comorbidities, those who are bed-ridden, along with those who needed a passive

change in position, chest percussion, rehabilitation, and mobilization. The top three procedures usually are done repeatedly. It might lead to the impression of severe and persistent pain in those patients. Favre et al. found that certain procedures (muscle strengthening, mobilization, weight-bearing, stretching, installation in bed, dressing-undressing) were common causes of care-related pain during rehabilitation after orthopedic trauma [18]. Caring for the elderly is a global issue. We will need to take care of our geriatric patients more often in the future as their numbers increase. Health caregivers, in hospitals or homes, should be careful with regards to the care-related pain surrounding rehabilitation activities.

We found that surgery was a factor significantly related to severe pain (Table 1), but not to severe pain over a long duration (Table 2). We have a few programs to shorten the pain duration. First, we have a pain status dashboard, which provides real-time pain score on our Electronic Hospital Information system. Second, our standard of operation requires in-charged nurses to check the pain status every 8 hours and respond to the breakthrough pain immediately. Third, the patients will be re-visited 30min and 60 min once they receive intravenous or oral pain control medicine, respectively. Fourth, minimally invasive surgery and patient-controlled analgesia (PCA) are widely used in our hospital. Fifth, those patients with PCA are visited for pain control every 4 hours or as needed. The above programs probably reduced the severe persistent pain, especially for the patient receiving a surgical intervention.

## Limitations of the study

Certain limitations within the study should be noted. First, our study findings may be limited in generalizability, as the sample subjects came from a single medical center. However, it remains as crucial data in Taiwan, and also in Asia. Second, data from the interviews may underestimate the patients' pain prevalence. We trained our interviewers prior to starting the study in order to minimize any bias. Our data are also similar to the results of European studies. Third, the inherent limitations of memory recall may have influenced our findings. Fourth, we did not investigate the already present pain before admission. Our study represented the overall pain prevalence of hospitalized patients. In spite of the limitations, this study offers critical new insight into the pain characteristics of inpatients.

## Conclusion

Our study and the available literature have revealed that pain prevalence inpatients average at around 70%, (ranging from 52%~89.5%) amongst hospitalized patients. Most of the data originated from western countries, while several were reported from Asian countries. No data originated from Taiwan. Pain is an essential issue surrounding patient safety and healthcare quality. We discovered that pain prevalence decreased with an incremental increase in age, and that we should pay close attention to possible underestimation due to the poor cognition of geriatric patients. It was noted that surgery was associated with severe pain, and usually treated immediately in our hospital. What needs to be done is assuring the prevention of surgical related acute pain; for example, patient-controlled analgesia or use of the program titled Early Recovery After Surgery. It remains necessary that we continue to carefully monitor the situation of care-related pain, particularly as its related to mobilization or rehabilitation. Overall, a periodic survey of pain prevalence is necessary if achieving "Towards a Pain-Free Hospital" is the ultimate goal.

## Supporting information

**S1 File.**
(XLSX)

## Acknowledgments

We thank the interviewers: Yi-Ting Liao, Ren-Qi Chen, Yun-Tong Sun, Ren-Jyun Yan, Hui-Hsun Chang, and Ying-Jyun Shih. They followed protocol and did their best to complete all of the questionnaires within a limited time frame.

## Author Contributions

**Conceptualization:** Chieh-Liang Wu, Yin-Lurn Hung, Hui-Mei Huang, Chia-Hui Chang, Chih-Cheng Wu, Chih-Jen Hung, Te-Feng Yeh.

**Data curation:** Yin-Lurn Hung, Yan-Ru Wang, Hui-Mei Huang.

**Formal analysis:** Te-Feng Yeh.

**Methodology:** Chieh-Liang Wu, Yin-Lurn Hung, Yan-Ru Wang, Hui-Mei Huang, Chia-Hui Chang, Te-Feng Yeh.

**Project administration:** Chieh-Liang Wu, Yin-Lurn Hung.

**Supervision:** Chih-Jen Hung.

**Writing – original draft:** Chieh-Liang Wu, Chih-Cheng Wu.

**Writing – review & editing:** Chieh-Liang Wu, Chih-Cheng Wu, Chih-Jen Hung, Te-Feng Yeh.

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
