## [Decision Letter · Decision Letter 0]

14 Aug 2020

PONE-D-20-16670

Pain prevalence in hospitalized patients at a tertiary academic medical center: exploring severe persistent pain

PLOS ONE

Dear Dr. Wu,

Thank you for submitting your manuscript to PLOS ONE. After careful consideration, we feel that it has merit but does not fully meet PLOS ONE’s publication criteria as it currently stands. Therefore, we invite you to submit a revised version of the manuscript that addresses the points raised during the review process.

We look forward to receiving your revised manuscript.

Kind regards,

Yan Li

Academic Editor

PLOS ONE

Journal Requirements:

2)  Thank you for including your ethics statement:  "The ethical committee/institutional review board (Institutional Review Board (II)107-B-08 Board Meeting) approved the study protocol (Protocol no./IRB, TCVGH No: CE18236B).".   

3) Please include captions for your Supporting Information files at the end of your manuscript, and update any in-text citations to match accordingly. Please see our Supporting Information guidelines for more information: http://journals.plos.org/plosone/s/supporting-information.

Reviewers' comments:

Reviewer's Responses to Questions

**Comments to the Author**

1. Is the manuscript technically sound, and do the data support the conclusions?

Reviewer #1: Yes

2. Has the statistical analysis been performed appropriately and rigorously? 

Reviewer #1: Yes

3. Have the authors made all data underlying the findings in their manuscript fully available?

Reviewer #1: Yes

4. Is the manuscript presented in an intelligible fashion and written in standard English?

Reviewer #1: Yes

5. Review Comments to the Author

Reviewer #1: Dear authors, here you receive my review regarding the manuscript entitled “Pain prevalence in hospitalized patients at a tertiary academic medical center: exploring severe persistent pain” with Manuscript Number: PONE-D-20-16670 and short Title: Pain prevalence in hospitalized patients. The authors present the results of a pain prevalence study based on questionnaire divided in 4 parts with special focus on demographics, pain status, pain management, and overall patient satisfaction. The study was performed within 19 wards of a tertiary referral medical center in Taiwan. The author remark on the fact that pain and pain management is not that well studied yet in some parts of Asia, in comparison with the literature in e.g. Europe.

The manuscript is well written with clear understandable English language. The length is acceptable. The study was approved by an ethical review board and the number is provided.

Besides the prevalence also the causes of care-related pain were surveyed.

However, there are some questions remaining.

The researchers seem to be trained in the execution of the questionnaire. This may be very important and could have had influence in the observed pan prevalence? We know for instance from previous studies that estimation of pain severity can be done by the patient and second best is the estimation by the responsible attending nurse taking care of the patient, when patients are unresponsive. (See, e.g., SJGM Ahlers , et al in Crit Care . 2008;12(1):R15.). What can be said about the training of the researchers?

Page 12 Line 23 “Questionnaire” the internal consistency reliability …. Excellent. What can be said or shown for the reader that this aspect was judged right?

P13 L18 what is meant by quota? Was there a limitation of the number of patients possible to enter the study? Were there any predetermined quota?

What was the minimum or maximum number of days in retrospect that the patients were referring to regarding the experienced pain during their hospital stay?

P14 L4 …longer than 4 hours…. How did the authors come to the discrimination of more or less than “4 hours”? Do they refer to any previous research or publications?

P15 Table 1 It is interesting to notice that patients with severe pain 28.9% and 31.6% of all patients were male or female , respectively. Maybe not significantly different?

Furthermore, when pooling the number of patients with ages from 40 till 80 31.6% of patients experienced severe pain in comparison with 30% of patients aged≤39 years of age

532 from 754 patients = 70% aged 40 to 80 years of age experienced pain.

The authors found that surgery related pain was one of the serious causes of pain among this high number of patients with any pain experience, regardless the seriousness. Is i possible to be more specific on the type of surgery or type of wound pain the patients experienced with mobilization.

Were there any [patients with non-surgery related pain and causes such as related to rheumatic bone and joint disease or oncology and did these numbers or prevalence differ between age groups?

Interesting in table 2 page 17 is that overall 21% of patients all ages together experience a NRS4-10 for longer than 4 hours.

P18 Table 3, what may be concluded or commented o “needle pain” Is this once performed in patients or in one patient repeatedly. Are there any signs that repeatedly performed panful procedures influenced the severity of pain longitudinally?

P19 “Discussion” The authors refer to the study of Fabbian et al with a prevalence of 7.% of patients with severe. Please try to improve your comment regarding your observation of approx.. 21% prevalence (pooled ages), which is extremely higher? Severe pain ages <39 years old of 30% and ages 40-80 was 31.6. This differs from the reference 9.

How many patients had already pain before hospitalization, for how long a period and was pain the main reason for admittance to the hospital? Did this influence your results?

This may be important as pain experienced in the hospital, as a result of the treatment, diagnostic procedures as well as mobilization, superimposed on already present pain may be more severe within the patient with an already triggered pain system.

P 20 L1 the word pan must be changed into pain

P20 L8 Cognitive impairment….express their pain. This seems very important to me. What could be concluded from this feature in your cohort, what is the influence on patient report. This also refers to the lines 5 and 6 on p22.

P19 and 20 We found…., but not a factor (Table2). Please explain? What is a possible explanation? Is there e.g. an influence from perioperative surgery and anaesthesia driven perioperative pain protocols or pain management programmes during the first 24 hours after surgery

6. PLOS authors have the option to publish the peer review history of their article (what does this mean?). If published, this will include your full peer review and any attached files.

Reviewer #1: **Yes: **dr Peter Bruins, MD, PhD, EDIC

---

## [Author Response · Author response to Decision Letter 0]

22 Sep 2020

Response to reviewers

PONE-D-20-16670

Pain prevalence in hospitalized patients at a tertiary academic medical center: exploring severe persistent pain

Academic editor comment: 

Thank you for including your ethics statement: "The ethical committee/institutional review board (Institutional Review Board (II)107-B-08 Board Meeting) approved the study protocol (Protocol no./IRB, TCVGH No: CE18236B).". Please amend your current ethics statement to include the full name of the ethics committee/institutional review board(s) that approved your specific study. Once you have amended this/these statement(s) in the Methods section of the manuscript, please add the same text to the “Ethics Statement” field of the submission form (via “Edit Submission”).

Answer: The full name of the IRB board is "Institutional Review Board I & II of Taichung Veterans General Hospital" The full name has been amended in the method section (page 6, Line 9-10) and also Ethics Statement field. 

Reviewers' comments:

1. Is the manuscript technically sound, and do the data support the conclusions? The manuscript must describe a technically sound piece of scientific research with data that supports the conclusions. Experiments must have been conducted rigorously, with appropriate controls, replication, and sample sizes. The conclusions must be drawn appropriately based on the data presented.

Reviewer #1: Yes

Answer: Thanks for the reviewer’s comment. 

2. Has the statistical analysis been performed appropriately and rigorously?

Reviewer #1: Yes

Answer: Thanks for the reviewer’s comment. 

3. Have the authors made all data underlying the findings in their manuscript fully available? The PLOS Data policy requires authors to make all data underlying the findings described in their manuscript fully available without restriction, with rare exception (please refer to the Data Availability Statement in the manuscript PDF file). The data should be provided as part of the manuscript or its supporting information, or deposited to a public repository. For example, in addition to summary statistics, the data points behind means, medians and variance measures should be available. If there are restrictions on publicly sharing data—e.g. participant privacy or use of data from a third party—those must be specified.

Reviewer #1: Yes

Answer: Thanks for the reviewer’s comment. 

4. Is the manuscript presented in an intelligible fashion and written in standard English? PLOS ONE does not copyedit accepted manuscripts, so the language in submitted articles must be clear, correct, and unambiguous. Any typographical or grammatical errors should be corrected at revision, so please note any specific errors here.

Reviewer #1: Yes

Answer: Thanks for the reviewer’s comment. 

5. Review Comments to the Author

Question 1: 

The researchers seem to be trained in the execution of the questionnaire. This may be very important and could have had influence in the observed pan prevalence? We know for instance from previous studies that estimation of pain severity can be done by the patient and second best is the estimation by the responsible attending nurse taking care of the patient, when patients are unresponsive. (See, e.g., SJGM Ahlers , et al in Crit Care . 2008;12(1):R15.). What can be said about the training of the researchers?

Answer: The role of the interviewers in the study is different from the role of the researchers in the reference (SJGM Ahlers, et al. in Crit Care. 2008;12(1):R15), that assessed the pain of the patients. Some of our patients were not able to fill the questionnaire themselves on a mobile IPAD. Therefore, we trained the interviewers to help the patients understand the questions and input their answers into IPAD. All of the interviewers had the background of healthcare administration and finished the 3 hours courses, including the content of the questionnaire and interview technique. The patients answered the questions of the questionnaire themselves. The interviewers imputed the answers into the web-based questionnaire site via a mobile IPAD.

We changed the sentences on page 7, Line 21, and 22 "The patients answered the questionnaire themselves. The interviewers imputed the patients’ answers into the web-based questionnaire site via a mobile IPAD.”

Question 2: 

Page 12, Line 23 "Questionnaire" the internal consistency reliability …. Excellent. What can be said or shown for the reader that this aspect was judged right?

Answer: The questionnaire is composed of 36 questions, divided into four main parts. The first part asks for demographic data, including age, gender, education, weight, disease categories, and surgical history. The second part evaluates pain status, the types of care-related pain, and a patient's literacy regarding pain. The third part is to define their experience surrounding pain management from the physicians and nurses. And the final part is to measure the patient's overall satisfaction. This study's materials were the first two parts, and internal consistent reliability (Cronbach’ α) was the content of the other parts. 

We change the sentence on page 6; line 22, "Five experts reviewed the questionnaire with a Content Validity Index (CVI) of 0.90. The study included the results of part 1 and 2 of the questionnaire.”

Question 3: 

P13 L18 what is meant by quota? Was there a limitation of the number of patients possible to enter the study? Were there any predetermined quota?

What was the minimum or maximum number of days in retrospect that the patients were referring to regarding the experienced pain during their hospital stay?

Answer: In order to consider budget and representativeness at the same time, this study used the number of beds in general wards of each nursing unit as a quota. The sample size of each nursing unit was the number of beds, the cases in each nursing unit would stop when they reached the quota. 

We added the sentence on page 7, lines 14 and 15, "The number of beds of each general ward was the quota, and we stopped enrollment if the quota was reached.”

Questions 4:

P14 L4 …longer than 4 hours…. How did the authors come to the discrimination of more or less than “4 hours”? Do they refer to any previous research or publications?

Answer: In terms of longer than 4 hours, we do not refer to any previous research or publication. Our nurses have to respond to the breakthrough pain and re-asses within one hour based on our SOP of pain management. Our study team of "Toward pain-free hospital" defined pain longer than 4 hours is inadequate management of pain control. 

We added the sentence to describe the persistent pain longer than 4 hours in page 7, Line 7-9 “Based on our standard of operation of “Toward a Pain-Free Hospital”, we have to respond the breakthrough pain within one hour and then define the duration of pain longer than 4 hours is persistent pain.” 

Question 5: 

P15 Table 1 It is interesting to notice that patients with severe pain 28.9% and 31.6% of all patients were male or female, respectively. Maybe not significantly different? 

Answer: It is significantly different statistically with p = 0.694

Question 6: 

Furthermore, when pooling the number of patients with ages from 40 till 80 31.6% of patients experienced severe pain in comparison with 30% of patients aged≤39 years of age. 532 from 754 patients = 70% aged 40 to 80 years of age experienced pain. The authors found that surgery related pain was one of the serious causes of pain among this high number of patients with any pain experience, regardless the seriousness. Is i possible to be more specific on the type of surgery or type of wound pain the patients experienced with mobilization.

Answer: We further grouped the types of surgery into the followings: brain (N= 22), head and neck (N=33), chest and cardiovascular (N=44), abdomen (N=102), retroperitoneal (N=32), gynecology (N=32), spine (N=55), extremities(N=44), debridement (N=94) and others. The above surgery types were not statistically associated with severe pain, moderate/severe pain, and moderate/severe pain with a duration longer than 4 hours. Based on the present data, we could not find the association between severe pain and surgery types. Further study is necessary to answer the question. So, we did not make any changes to the comment in the manuscript. 

Question 7: 

Were there any [patients with non-surgery related pain and causes such as related to rheumatic bone and joint disease or oncology and did these numbers or prevalence differ between age groups?

Answer: This study was a hospitalized survey with quota sampling in each ward. We did not focus on a specified disease, for example, rheumatic bone and joint disease or oncology. It is difficult to answer the above comment. However, we try our best to define the cases with cancer-based on their diagnosis at discharge. We did not find the association of severe pain, moderate/severe pain, and moderate/severe pain with longer than 4 hours significantly. So, we did not make any changes to the comment in the manuscript. 

Question 8: 

Interesting in table 2 page 17 is that overall 21% of patients all ages together experience a NRS4-10 for longer than 4 hours.

Answer: There are two errors in the number. 

In Table 2 (the row of case number), pain � 4 h “N=312” was corrected as N=177 and NRS 4-10 pain � 4 “N=177” was corrected as N=143. In the table 2 on page 11. 

Overall, around 20% (143/719) of patients experienced the pain with NRS 4-10 and longer 4 hours simultaneously. 

Question 9: 

P18 Table 3, what may be concluded or commented o “needle pain” Is this once performed in patients or in one patient repeatedly. Are there any signs that repeatedly performed panful procedures influenced the severity of pain longitudinally?

Answer: Thanks for the reviewer's comment. We agree your comments. Needle pain, wound dressing pain and change posture/chest percussion are all be done repeatedly. Those procedures might lead to severe and persistent pain. We added the sentence into the discussion on page 16, Line 14-15, "The top three procedures usually are done repeatedly. It might lead to the impression of severe and persistent pain in those patients.” 

Question 10:

P19 “Discussion” The authors refer to the study of Fabbian et al. with a prevalence of 7.% of patients with severe. Please try to improve your comment regarding your observation of approx.. 21% prevalence (pooled ages), which is extremely higher? Severe pain ages <39 years old of 30% and ages 40-80 was 31.6. This differs from the reference 9.

Answer: As the reviewer comments, our cohort has a high prevalence of severe pain. Our study's setting was different from the study of Fabbian, which enrolled the patients of the 30-bed internal medicine ward. We enrolled the patients from 19 general wards, under surgical or medical service. Our hospital is also a medical center with a high case mixed index (CMI) up to 1.3. Also, nearly half of the patients received operation. Operation is also a significant risk factor for severe pain. Therefore, the high percentage of patients with severe pain is possible. 

We revised the discussion on page 13, line 21-25. “However, we noted that the percentage of patients with severe pain was up to 30% (312/1034), which was much higher than the findings of Rabbian et al. [9]. Our hospital is a 1500-bed tertiary academic medical center and takes care of the patients with high severity (Case Mixed index 1.3 in 2018). Nearly half of the enrolled patients received the surgical intervention (Table 1). Therefore, a high percentage of severe pain seems rational in the study.”

Question 11: 

How many patients had already pain before hospitalization, for how long a period and was pain the main reason for admittance to the hospital? Did this influence your results?

This may be important as pain experienced in the hospital, as a result of the treatment, diagnostic procedures as well as mobilization, superimposed on already present pain may be more severe within the patient with an already triggered pain system.

Answer: In the study design, we focused on the prevalence of pain among hospitalized patients in general wards. The questionnaire did not set the questions to explore the already present pain before admission. 

We added this to the study limitation in the discussion on page 17, line 12-14 "Fourth, we did not investigate the already present pain before admission. Our study represented the overall pain prevalence of hospitalized patients. Despite the limitations, this study offers critical new insight into the pain characteristics of inpatients.” 

Question 12 

P 20 L1 the word pan must be changed into pain

Answer: The type error was corrected as “pain” on page 15 Line 5. 

We also corrected the type error "the", which was corrected as "they" on page 15, Line 2

Question 13

P20 L8 Cognitive impairment….express their pain. This seems very important to me. What could be concluded from this feature in your cohort, what is the influence on patient report. This also refers to the lines 5 and 6 on p22.

Answer: We did not have the concrete data in the cohort to support the concept, "Cognitive impairment in the geriatric population may limit their ability to express their pain." As the literature from Rottenberg et al. (Reference 14), aged patients were more common in poor self-rated health. 

We revised the sentence as “Poor self-rated health was more common in advanced aged patients [14]. Thee geriatric population may have limited ability to express their pain.” Page 15 Line 12-13

Question 14 

P19 and 20 We found…., but not a factor (Table2). Please explain? What is a possible explanation? Is there e.g. an influence from perioperative surgery and anaesthesia driven perioperative pain protocols or pain management programmes during the first 24 hours after surgery

Answer: We have four programs for pain management. (1) We have a pain status dashboard, which provides real-time pain score on our Electronic Hospital information system. (2) Our standard of operation requires in-charged nurses to check the pain status every 8 hours and respond to the breakthrough pain immediately. (3) The patients will be re-visited 30min and 60 min once they receive intravenous or oral pain control medicine, respectively. (4) Minimally invasive surgery and patient-controlled analgesia (PCA) are widely used in our hospital. (5) Those patients with PCA are visited for pain control every 4 hours or as needed. The above programs maybe reduce severe pain for a long time. 

Therefore, we deleted the sentence “We found that surgery was a strong factor related to severe pain (Table 1), but not a factor significantly related to severe pain over a long duration (Table 2) in the paragraph of Chronic pain may superimpose on the pain of inpatients" on page 15. 

We added a paragraph about how to shorten the duration of pain in the discussion. From page 16, Line 21 to Page 17, Line 5, "We found that surgery was a factor significantly related to severe pain (Table 1), but not to severe pain over a long duration (Table 2). We have a few programs to shorten the pain duration. First, we have a pain status dashboard, which provides real-time pain score on our Electronic Hospital information system. Second, our standard of operation requires in-charged nurses to check the pain status every 8 hours and respond to the breakthrough pain immediately. Third, the patients will be re-visited 30min and 60 min once they receive intravenous or oral pain control medicine, respectively. Fourth, minimally invasive surgery and patient-controlled analgesia (PCA) are widely used in our hospital. Fifth, those patients with PCA are visited for pain control every 4 hours or as needed. The above programs probably reduced the severe persistent pain, especially for the patient receiving a surgical intervention.”

---

## [Decision Letter · Decision Letter 1]

24 Nov 2020

Pain prevalence in hospitalized patients at a tertiary academic medical center: exploring severe persistent pain

PONE-D-20-16670R1

Dear Dr. Yeh,

We’re pleased to inform you that your manuscript has been judged scientifically suitable for publication and will be formally accepted for publication once it meets all outstanding technical requirements.

Kind regards,

Yan Li

Academic Editor

PLOS ONE

Additional Editor Comments (optional):

Reviewers' comments:

Reviewer's Responses to Questions

**Comments to the Author**

1. If the authors have adequately addressed your comments raised in a previous round of review and you feel that this manuscript is now acceptable for publication, you may indicate that here to bypass the “Comments to the Author” section, enter your conflict of interest statement in the “Confidential to Editor” section, and submit your "Accept" recommendation.

Reviewer #2: (No Response)

2. Is the manuscript technically sound, and do the data support the conclusions?

Reviewer #2: (No Response)

3. Has the statistical analysis been performed appropriately and rigorously? 

Reviewer #2: (No Response)

4. Have the authors made all data underlying the findings in their manuscript fully available?

Reviewer #2: (No Response)

5. Is the manuscript presented in an intelligible fashion and written in standard English?

Reviewer #2: (No Response)

6. Review Comments to the Author

Reviewer #2: (No Response)

7. PLOS authors have the option to publish the peer review history of their article (what does this mean?). If published, this will include your full peer review and any attached files.

Reviewer #2: No

---

## [Editor Report · Acceptance letter]

27 Nov 2020

PONE-D-20-16670R1 

Pain prevalence in hospitalized patients at a tertiary academic medical center: exploring severe persistent pain 

Dear Dr. 葉:

I'm pleased to inform you that your manuscript has been deemed suitable for publication in PLOS ONE. Congratulations! Your manuscript is now with our production department. 

Kind regards, 

on behalf of

Dr. Yan Li 

Academic Editor

PLOS ONE